THE NATURAL HISTORY OF MODEL ORGANISMS

# Insights into the evolution of social systems and species from baboon studies

**Abstract** Baboons, members of the genus *Papio,* comprise six closely related species distributed throughout sub-Saharan Africa and southwest Arabia. The species exhibit more ecological flexibility and a wider range of social systems than many other primates. This article summarizes our current knowledge of the natural history of baboons and highlights directions for future research. We suggest that baboons can serve as a valuable model for complex evolutionary processes, such as speciation and hybridization. The evolution of baboons has been heavily shaped by climatic changes and population expansion and fragmentation in the African savanna environment, similar to the processes that acted during human evolution. With accumulating long-term data, and new data from previously understudied species, baboons are ideally suited for investigating the links between sociality, health, longevity and reproductive success. To achieve these aims, we propose a closer integration of studies at the proximate level, including functional genomics, with behavioral and ecological studies.
DOI: https://doi.org/10.7554/eLife.50989.001

**JULIA FISCHER[†]*, JAMES P HIGHAM[†]*, SUSAN C ALBERTS, LOUISE BARRETT, JACINTA C BEEHNER, THORE J BERGMAN, ALECIA J CARTER, ANTHONY COLLINS, SARAH ELTON, JOËL FAGOT, MARIA JOANA FERREIRA DA SILVA, KURT HAMMERSCHMIDT, PETER HENZI, CLIFFORD J JOLLY, SASCHA KNAUF, GISELA H KOPP, JEFFREY ROGERS, CHRISTIAN ROOS, CAROLINE ROSS, ROBERT M SEYFARTH, JOAN SILK, NOAH SNYDER-MACKLER, VERONIKA STAEDELE, LARISSA SWEDELL, MICHAEL L WILSON AND DIETMAR ZINNER**

*For correspondence: jfischer@dpz.eu (JF); jhigham@nyu.edu (JPH)

[†]These authors contributed equally to this work

**Competing interests:** The authors declare that no competing interests exist.

## Introduction

Humans have been captivated by baboons for thousands of years: from ancient Egypt, where the god of wisdom, Thoth, was depicted with a baboon head, to the mid-19th century when Charles Darwin remarked, "He who understands baboon would do more towards metaphysics than Locke" (*Darwin, 1838*). At the beginning of the 20th century, the South African naturalist Eugene Marais provided one of the first detailed accounts of free-ranging baboons (*Marais, 1939*), and by the 1950s, baboons had become the subject of more systematic scientific enquiry, both in the field and in captivity. This was the decade that the American physical anthropologist Sherwood Washburn and his student Irven DeVore set out to investigate baboons in Kenya (*Vore and Washburn, 1961*). Washburn reasoned that these ground-living primates were a good model for early human adaptations because they evolved in African savannas alongside ancestral hominins. Meanwhile, increasing interest in the use of non-human primates as biomedical models for humans led to the funding in 1958 of a three-year proposal titled "Initiation and Support of Colony of Baboons" by the US National Institutes of Health, with the first group of baboons shipped to the United States from Kenya in 1960 (*Vande-Berg, 2009*). Since then, research in captivity on

**Figure 1.** Distribution of the six *Papio* species. Species distributions are modified from *Zinner et al. (2013)*. Male baboon drawings by Stephen Nash. Reprinted with permission from *Fischer et al. (2017)*.
DOI: https://doi.org/10.7554/eLife.50989.003

baboons as a biomedical model has been complemented by extensive fieldwork on baboon populations across Africa. Knowledge of the links between health and fitness in baboons under natural circumstances, including natural levels of genotypic and phenotypic variation, appears critical to put results from captive studies into context. An understanding of the evolution and life history of these animals in the wild also allows the scientific community to assess the validity of results derived from captive populations.

While the earlier field studies set out to uncover a baboon archetype, subsequent research has revealed that there is no such thing as "the baboon". Indeed, many would argue that the value of this genus lies precisely in the substantial variation in the social systems, life histories and ecologies within and between the

baboon species (see *Box 1* for a glossary of specialist terms used in this article). Collectively, these characteristics make baboons an excellent model organism for investigating a range of fundamental biological processes, such as physiological and behavioral adaptation, hybridization and speciation with gene flow (*Alfred and Baldwin, 2015*). In this way, the baboon model constitutes an important complement to other mammalian model organisms, such as wild house mice(*Mus musculus*; *Phifer-Rixey and Nachman, 2015*) and deer mice (genus *Peromyscus*; *Bedford and Hoekstra, 2015*).

## Systematic classification and distribution

Within the primate order, all extant baboons belong to the genus *Papio*. The genus is part of the tribe Papionini, within the family

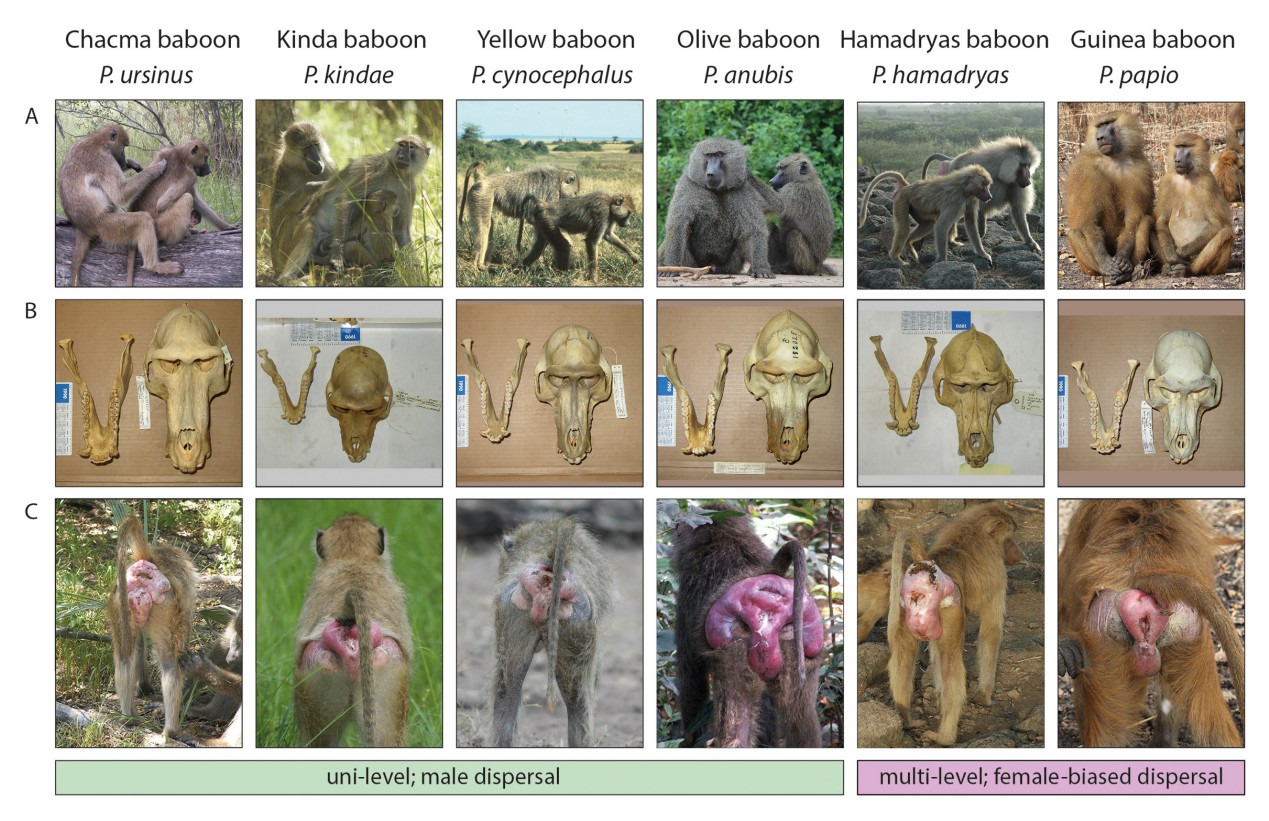

**Figure 2.** Illustration of key traits across baboon species. (A) Phenotypic variation between species. Pictures show adult males and females. (B) Crania of male baboons. (C) Sexual swellings of female baboon during peak estrus. Species are grouped by social organization (uni- and multi-level) and dispersal behavior (male- or female-biased dispersal). Images from Alexis Amann, Andrea Cardini, Sarah Elton, Julia Fischer, Courtney Fitzpatrick, James Higham, Megan Petersdorf, Joan Silk and Larissa Swedell.
DOI: https://doi.org/10.7554/eLife.50989.004

Cercopithecidae. The fossil record and phylogeographic analyses indicate that baboons originated in southern Africa. Nuclear and mitochondrial estimates put the date of initial divergence of baboon lineages at 1.5–2.1 million years ago (*Newman et al., 2004*; *Rogers et al., 2019*; *Zinner et al., 2013*). At about the same time, during the Pleistocene epoch, baboons started to expand their range across sub-Saharan Africa into both northern and southern savannas.

Presently, six species are recognized: the chacma baboon (*Papio ursinus*), which is found in southern Africa; the yellow baboon (*Papio cynocephalus*), which inhabits large parts of eastern Africa; the Kinda baboon (*Papio kindae*), which is found in Zambia, eastern Angola, and southern DR Congo; the olive baboon (*Papio anubis*), whose distribution ranges from northern DR Congo to parts of Kenya and Tanzania across to Sierra Leone in the west and to Eritrea in the east; the Guinea baboon (*Papio papio*), which is found from Sierra Leone to Mauritania and Senegal; and the hamadryas baboon (*Papio hamadryas*), which inhabits parts of Eritrea, Ethiopia, Somalia and the south-western part of the Arabian peninsula (*Figure 1*). Hybrid zones are found where species' distributions come into contact.

While the systematic grouping into taxa within the genus *Papio* is well accepted on both phenotypic and genetic evidence, the taxonomic ranking is disputed. According to the biological species concept (*Mayr, 1963*), all taxa would be considered one polytypic species (*Papio hamadryas*) because where they meet in the wild they interbreed freely, producing viable and fertile hybrid offspring. Given that the different taxa vary substantially in appearance, behavior, and the characteristics of their society, we follow the phylogenetic species concept (*Cracraft, 1983*), and refer to the different taxa as "species".

## Box 1. Glossary.

**Admixture**: Genetic admixture refers to the exchange of genetic information among two populations or taxa that had been reproductively isolated and which genetically diverged (see introgressive hybridization).

**Consortship**: Consortships occur when females are sexually receptive and involve a male and female pair who associate in close proximity, often mating repeatedly. Typically, male-female consort pairs travel, feed and rest together. Consortships can last for hours or days.

**Genetic architecture**: Refers to the underlying genetic basis of a phenotypic trait (morphological, physiological, behavioral) and the variation in the respective trait.

**Ghost lineage**: A term from paleontology and phylogenetics. It refers to a phylogenetic lineage that has no fossil record or living representatives, but is inferred to have existed, for example, by whole-genome analyses of related taxa.

**Hybridization**: The interbreeding between two differentiated populations, usually closely related species, resulting in the combination of genetic material from previously isolated gene pools.

**Introgression or introgressive hybridization**: Observed between species or between genetically well-separated populations. It refers to the movement of genes, or gene flow, from one species into the gene pool of another by the repeated backcrossing of interspecific hybrids with one of their parent species.

**Life history**: The life history of an organism is a characterization of its patterns of development, reproduction, aging and mortality. Key measures of primate life history include length of gestation, age at the first occurrence of menstruation, age at first reproduction, number of offspring per litter, number of births per year, interval between births and life span.

**Mating system**: The distribution of matings among sexually active individuals within a social unit. Primate species can be monogamous (mating occurs mostly between pair partners), polyandrous (one female mates with multiple males), polygynous (one male mates with multiple females), or polygynandrous (males and females have multiple mating partners).

**Mitochondrial and nuclear lineages**: Mitochondria, organelles of almost all eukaryotic cells, carry their own genome. In contrast to nuclear genomes, recombination of the mitochondrial genome is a rare event and, since mitochondria are almost exclusively inherited via the maternal lineage, nuclear and mitochondrial genetic lineages can experience independent evolutionary histories. This often results in discordant phylogenies when using sex chromosomes (gonosomal), non-sex chromosomes (autosomal) or mitochondrial markers. Even phylogenies based on different nuclear genes or parts of the nuclear genome can lead to some discordances. Nevertheless, one can use nuclear and mitochondrial lineages to infer different evolutionary events within the evolutionary history of a species.

**Phylogenetic species concept**: This concept defines a species as an irreducible group or cluster whose members are descendants from a common ancestor and who all possess a combination of certain defining derived traits known as apomorphies. Such groups are monophyletic (contrasted with paraphyletic or polyphyletic groups). Reproductive isolation is not a precondition for the definition of species. Since monophyletic groups are often nested, ranking a particular group as a species can be problematic.

**Social organization**: The number of individuals and the composition of a group, including when and where those individuals are distributed. Groups may for instance be stable or reveal a fission-fusion system where the group temporarily splits into smaller sub-groups. Baboon societies may be uni-level (individuals live in a stable group and generally roam together) or multi-level (groups consist of predictable sub-groups, which may in turn consist of smaller sub-groups). An important aspect of the social organization is the dispersal behavior, that is, which sex typically remains in the group into which is was born (i.e. its natal group) and which sex leaves the natal group to breed. In most mammals, females stay (female philopatry) and males leave (male dispersal), which is considered the ancestral state. The timing and type of dispersal has important consequences for the genetic structure of groups.

**Social style**: The degree of aggressiveness among group members in a species. In societies that exhibit steep dominance hierarchies ("high despotism"), aggression is extremely asymmetrical, while in tolerant species, aggression is mild and frequently bi-directional. A further important component is the degree of nepotism within the species, that is, how kin-biased affiliation is. The social style of a species is associated with variation in relationship quality, which in turn characterizes the social structure of a species.

**Social system**: A primate species' social system encompasses its social organization, social style, mating patterns and parental care system (*Kappeler, 2019*).

DOI: https://doi.org/10.7554/eLife.50989.002

## Morphology

All baboon species are anatomically and morphologically well adapted to a quadrupedal terrestrial lifestyle (*Fleagle, 2013*). They have dog-like muzzles and males have large canine teeth. Depending on the species, body mass varies between 17 and 30 kg for adult males, and between 10 and 15 kg for females, resulting in a sexual dimorphism in mass ranging between 1.55 and 2.20 (*Anandam et al., 2013*; *Fischer et al., 2017*; *Swedell, 2011*).

The species differ notably in their fur color, body size and sexually selected characteristics, such as the distinct capes which are most pronounced in Guinea and hamadryas baboons but also present in olive baboons. Females of all species develop sexual swellings of the anogenital region when they are fertile. These swelling change throughout the menstrual cycle, such that maximum swelling typically coincides with ovulation (*Higham et al., 2008*). The size and shape of the swellings varies considerably among species (*Figure 2*; *Petersdorf et al., 2019*).

## Ecology

All baboon species are largely terrestrial during the day but retreat to sleeping trees or cliffs during the night. They exhibit great ecological flexibility, allowing them to occupy habitats including semi-deserts grasslands, woodland savannas, humid forests, and Afroalpine grasslands over 3,000 meters above sea level (*Chala et al., 2019*; *Fuchs et al., 2018*). Baboons eat a broad range of foods, although their diet mainly consists of plants, including fruit, seeds, leaves, and roots. They also eat insects and other arthropods and, occasionally, kill small antelopes, hares, rodents, birds and smaller monkeys (*Goffe and Fischer, 2016*; *Swedell, 2011*).

## Phylogeography

Phenotypic differences between species are well recognized (*Jolly, 1993*), and based on their molecular phylogeny, baboons are generally split in two major groups: north and south (*Dunn et al., 2013*; *Frost et al., 2003*). However, genetic evidence reveals a complex evolutionary history of the genus *Papio*. Analysis of mitochondrial DNA yields a phylogeny that includes several major haplogroups or clades – groups of individuals who belong to a specific mitochondrial lineage. These haplogroups reflect the geographic origin of the respective specimens better than their external phenotypes or taxonomic classification, making species appear to be paraphyletic and polyphyletic when mapped onto the mitochondrial phylogeny (*Zinner et al., 2009*; *Zinner et al., 2011*).

Comparisons of whole genome sequences confirm the six baboon species taxonomy and suggest that the initial evolutionary divergence separated a southern lineage that ultimately produced Kinda, chacma and yellow baboons, from a northern lineage that produced olive, hamadryas and Guinea baboons (*Rogers et al., 2019*). Ancient hybridization events appear to have affected the genetic makeup of all species. For instance, Guinea baboons most likely experienced genetic admixture with a "ghost lineage" that is probably extinct, or that is at least not represented in the sample of genomes analyzed to date (*Rogers et al., 2019*).

Multiple episodes of admixture and introgression have been linked to climate change and range expansion (*Rogers et al., 2019*; *Walker et al., 2017*; *Wall et al., 2016*; *Zinner et al., 2013*). Similar evolutionary mechanisms, including gene transfer by introgressive hybridization, are now recognized to have influenced the evolution of Neanderthals, Denisovans and modern humans (*Green et al., 2010*; *Prüfer et al., 2017*; *Prüfer et al., 2014*; *Reich et al., 2010*; *Ackermann et al., 2019*). However, the absence of genomic data from

non-sapiens African hominins presently hinders our ability to ask questions about ancestral African hominin hybridization (*Scerri et al., 2018*; *Stringer, 2016*). Baboons allow us to study the impact of gene flow in an extant model.

Of particular interest for understanding baboon evolution is how changes in population density and spatial structure, such as the opening and closing of forest and other barriers, gave rise to different social systems (*Jolly, 2019*). The range expansion of the genus appears to be of particular relevance. Given a southern African origin, modern baboons experienced a tremendous expansion of their range, possibly linked to changes to the habitats, animal communities and climate that occurred during the Pleistocene and that gave baboons the chance to disperse into the savanna belt north of the tropical forest zone (*Dolotovskaya et al., 2017*; *Zinner et al., 2011*). Like humans and other savanna species, baboons have thus been subject of recurrent range shifts, fragmentation, and isolation and reconnection of populations (*Zinner et al., 2011*) – dynamics that affected baboon genetic structure and speciation (*Rogers et al., 2019*).

In summary, baboons can serve as a valuable model for evolutionary divergence and hybridization, triggered by climatic changes and the expansion and fragmentation of populations in the African savanna. Such analyses are also highly relevant for a better understanding of early hominins.

## Variation in social organization and behavior

The six baboon species vary substantially in their social characteristics, including social organization, social style and mating patterns. Group size varies within and among species. In chacma, olive and yellow baboons, group size ranges from about a dozen up to roughly one hundred animals (*Markham et al., 2015*; *Swedell, 2011*), while hamadryas and Guinea baboons temporarily aggregate into groups of several hundreds of individuals (*Patzelt et al., 2011*; *Swedell, 2013*). Sex ratios in baboons vary too; some groups are fairly balanced, while adult females in other groups can outnumber adult males by about 10 to 1 (*Swedell, 2011*).

Chacma, olive, Kinda and yellow baboons – recently dubbed "COKY" baboons (*Jolly, 2019*) – live in multi-male-multi-female groups, in which related females constitute the stable core, while males leave the group they were born into and join another. Clear rank hierarchies among males

and females can be discerned based on aggressive interactions, including threats, chases and physical aggression, as well as signals of submission. In females, related individuals (known as matrilines) typically occupy adjacent ranks. For female chacma, olive and yellow baboons, female kin constitute the most important social partners (*Silk et al., 2017*; *Silk et al., 2010*; *Silk, 2003*). In Kinda baboons, however, males are the most significant grooming partners for females (*Petersdorf et al., 2019*).

Females of all COKY species interact and mate with several males in the group. High-ranking males generally experience higher mating success than lower-ranking ones, though this rank-related mating skew is more pronounced in chacma baboons than in olive or yellow baboons (*Bulger, 1993*; *Henzi and Barrett, 1999*; *Städele et al., 2019*). During female receptive periods, males aggressively guard their female mating partner, resulting in sexual "consortships" (*Noë and Sluijter, 1990*; *Smuts, 1985*). Consorts may last from several hours up to several days. Consort success (and thus mating success) is often related to male dominance status (*Gesquiere et al., 2011*). In yellow and olive baboons, however, male coalitions may be able to take the female away from a dominant male (*Noë and Sluijter, 1990*; *Smuts, 1985*).

Male competition and aggressiveness vary considerably between species. Infanticide is frequent in some populations of chacma baboons (*Palombit et al., 2001*), but rare in olive, yellow and hamadryas baboons (*Lemasson et al., 2008*; *Swedell, 2011*). Lactating females often form close ties to specific males, which are often the sires of their infants (*Huchard et al., 2010*; *Moscovice et al., 2010*; *Nguyen et al., 2009*; *Städele et al., 2019*). These relationships appear to be an adaptation against infanticide by recent immigrant males (*Palombit et al., 1997*) and harassment by other group members (*Alberts et al., 2003*) as well as a form of paternal investment (*Buchan et al., 2003*; *Huchard et al., 2013*). Male alliances are absent in chacma baboons, while common in yellow and olive baboons (*Noë and Sluijter, 1995*). These differences in male competitive regimes are reflected in their dispersal behavior: male chacma baboons in the Okavango delta, for instance, do not emigrate from their natal group until after they are fully grown (*Beehner et al., 2009*), while male olive and yellow baboons often emigrate during adolescence (*Alberts and Altmann, 1995*; *Packer et al., 1995*).

Over the past decade, studies of Kinda baboons have broadened our knowledge of morphological and behavioral variation within the genus. Kinda baboons are smaller in body size, have reduced sexual dimorphism in body and canine size, and have larger relative testis volume, compared to other baboon species (*Jolly, 2019*). Kinda females exhibit small sexual swellings (*Figure 2*) and give inconspicuous calls (*Petersdorf et al., 2019*). Chacma females, in contrast, give loud copulation calls, which function to incite male-male competition (*O'Connell and Cowlishaw, 1994*).

Hamadryas baboons – in contrast to the COKY baboon species described above – live in a multi-level society with reproductive units, called "one male units" comprising one sexually active leader male, a variable number of females, and sometimes a follower male (*Kummer, 1968*). Associations between several one-male units constitute a clan (*Abegglen, 1984*); several clans and unaffiliated bachelor males form a band, the main ecological unit, and multiple bands coalesce at resources, especially sleeping sites, to form troops (*Schreier and Swedell, 2009*). Recent behavioral and genetic studies of hamadryas baboons show that leader and follower males tend to be maternally related, in line with the fact that they disperse only rarely. Females within a unit are also more likely to be related than expected by chance (*Städele et al., 2016*).

Guinea baboons also live in a multi-level society. Several units comprising a primary male, 1–6 females, young, and occasional secondary males make up parties, and 2 to 3 parties constitute a gang within a larger community (*Fischer et al., 2017*). Male Guinea baboons maintain strong bonds and a high degree of spatial tolerance (*Fischer et al., 2017*). Some, but not all males with strong bonds are highly related, suggesting that the existence of kin in the group promotes male tolerance (*Patzelt et al., 2014*). In striking contrast to most other baboon species, aggression between males is so rare that it is not possible to discern a dominance hierarchy with certainty (*Dal Pesco and Fischer, 2018*). Males engage in extended ritualized greetings that apparently function to reinforce delineations between parties and to test bonds between males (*Dal Pesco and Fischer, 2018*). Females freely transfer between units, parties and gangs. Female tenure with a given male may vary between weeks and years (*Goffe et al., 2016*). Both Guinea and hamadryas baboons exhibit female-biased dispersal (*Kopp et al., 2015*; *Städele et al., 2015*).

Note that many of the most significant differences in social behavior between species have been observed across different populations in multiple African sites, as well as in captivity. Thus, there is good evidence that the variation we describe here reflects true species differences and not just variation between populations. Yet, characterizing the variation within species in greater detail would be extremely valuable.

Despite the variation in social organization and aggressiveness between the different baboon species, there is very little variation in the vocal repertoires and call types within the genus (*Hammerschmidt and Fischer, 2019*). This suggests that the structure of vocal patterns is highly conserved. Because species vary in their aggressiveness and their propensity to affiliate, they also differ in the frequency with which they use signals that either relate to fighting ability or "benign intent", respectively (*Faraut et al., 2019*; *Fischer et al., 2017*).

## Variation in social cognition

Variation in social organization and in the nature and extent of competition over resources between baboon species is thought to result in differential selective pressure on social cognition (*Amici et al., 2008*; *Aureli et al., 2008*). To date, most of the work on baboon social knowledge has been done on chacma baboons that exhibit steep dominance hierarchies (known as despotism). A suite of studies by the American primatologists Dorothy Cheney and Robert Seyfarth and colleagues revealed that chacma baboons have sophisticated social knowledge (reviewed in *Cheney and Seyfarth, 2008*). For instance, the animals represent the nested hierarchical rank relationships of their group members (*Bergman, 2003*), track the consortship status of pairs in their group (*Crockford et al., 2007*), and selective deploy aid to unrelated individuals that were former grooming partners (*Cheney et al., 2010*).

Field playback experiments revealed that baboon species respond differently to social information. While the territorial chacma baboons respond strongly to apparent intruders (*Kitchen et al., 2013*), the spatially tolerant Guinea baboons paid more attention to vocalizations from co-resident group members compared to neighbors or strangers (*Maciej et al., 2013*). Similarly, chacma baboons respond strongly to simulated rank reversals

(*Bergman, 2003*) or break-ups of existing consortships (*Crockford et al., 2007*), while Guinea baboon males were more interested in social information consistent with current social association patterns (*Faraut and Fischer, 2019*). The somewhat surprising responses of the Guinea baboons may be a result of the high gregariousness of the species, where deviant interaction patterns may initially be classified as "social noise" (*Faraut and Fischer, 2019*). In summary, these findings suggest that the content of what is represented, namely the associations between different individuals or their group memberships, appears to be relatively similar across the two species, while the value of different types of social information may vary substantially in relation to the type of society.

## Sociality, health, aging and fitness
Over the past decade, baboon research has provided ground-breaking insights into the relationships between social status, social relationships, health and fitness measures such as offspring survival and longevity. Data from two long-term studies of baboon behavior and life history suggest that sociality enhances the fitness of females. For example, infants born to yellow baboon females who are more socially integrated have higher survival than infants of less social mothers (*Archie et al., 2014*; *Silk, 2003*); similar patterns are also seen in chacma baboons (*Silk et al., 2010*). As in many other primates, higher-ranking male baboons sire more offspring than other males (*Altmann et al., 1996*). Higher-ranking females have shorter periods before they resume menstrual cycling following birth (*Gesquiere et al., 2018*; *Johnson, 2003*; *Packer et al., 1995*; *Smuts and Nicolson, 1989*; *Wasser et al., 2004*), which may be linked to quicker restoration of positive energy balance (*Gesquiere et al., 2018*). Consistent with this, feeding on crops in olive baboons (*Higham et al., 2009*), or discarded human food for yellow baboons (*Altmann et al., 1977*), also leads to a quicker return to menstrual cycling and increases reproductive output.

A number of studies have investigated the proximate mediators of the relationship between behavior and fitness. In particular, many researchers have taken advantage of non-invasive ways to measure glucocorticoid hormones, a class of hormones known to mediate the energetic demands that accompany social and ecological challenges. Concentrations of glucocorticoid hormones increase during specific challenges that are known to threaten an individual's fitness. For example, lactating chacma females that were at risk for infanticide because a new male immigrated into the group exhibited elevated glucocorticoid hormones compared to female counterparts that were not at risk (*Beehner et al., 2005*). Additionally, loss of a close female relative increases glucocorticoid concentrations, and this increase may be responsible for initiating a compensatory broadening and strengthening of female grooming networks (*Engh et al., 2006*).

Several studies have investigated the relationship between glucocorticoid concentrations, rank and social stability in male baboons. In a long term-study of yellow baboons, high-ranking males had lower glucocorticoid concentrations, regardless of hierarchy stability, while alpha males may experience higher concentrations than expected for their rank (*Gesquiere et al., 2011*). Nonetheless, high-ranking yellow baboon males get sick less often and heal from wounds faster, suggesting that these high-ranking males are in better health and do not suffer trade-offs from these extra demands (*Archie et al., 2012*). Higher-ranking chacma baboon males also had higher glucocorticoid concentrations (*Bergman et al., 2005*; *Kalbitzer et al., 2015*).

More recently, baboons were also established as a promising model for studying the impact of sexually transmitted diseases on mating behavior. Female olive baboons in a population infected with the bacterium *Treponema pallidum*, a substrain of which causes syphilis in humans, copulated less with males showing clinical signs of infection (*Paciência et al., 2019*). These findings highlight how pathogens may impose important selective pressures in mate choice and ultimately social evolution.

## Functional genomics
The addition of data on Kinda and Guinea baboons increases the value of the baboon as a model, as we begin to have data available for all baboon species. While one aim of future analyses will be to understand the sources of variation between species, documenting similarities is equally valuable. Technological developments in genomic sequencing (*Robinson et al., 2019*; *Rogers et al., 2019*), including from non-invasively collected samples such as feces (*Perry et al., 2010*; *Snyder-Mackler et al., 2016*), have brought genomics to the forefront of baboon behavioral studies (*Tung et al., 2010*). Given the close phylogenetic relatedness

of the six baboon species, variation in key aspects of social behavior, and the presence of hybrids displaying intermediate phenotypes within hybrid zones, investigation of causal pathways from genotype to phenotype seems particularly promising within the baboon model (*Bergey et al., 2016*; *Jolly et al., 2008*).

Formerly, research into primate behavioral genetics focused on identifying a few small specific functional polymorphisms in sequence or length, and on linking these to phenotypic variation (e.g., *Kalbitzer et al., 2016*). However, such studies are likely to overestimate the effect of one single aspect of genetic variation. With genotyping of single nucleotide polymorphisms (SNPs), and whole-genome sequencing, primate field studies are beginning to explore the wider genomic architecture that underlies variation in social behavior (*Rogers, 2018*). As well as whole genome sequence data, researchers now have access to annotations for protein coding genes and transcriptomes (*Robinson et al., 2019*; *Rogers et al., 2019*; *Vilgalys et al., 2019*). We therefore expect an exponential increase in the number and diversity of available genomes, which will facilitate the investigation of the basis of baboon adaptations and adaptive flexibility. In conjunction with research on other model organisms, such as deer mice (*Bedford and Hoekstra, 2015*), such studies provide fundamental insights into the foundation of natural variation and adaptation in socially living mammals.

## Box 2. Outstanding questions about the natural history of baboons.

- How have changes in population density and environmental conditions (e.g., opening and closing of forest and other barriers) affected dispersal and mating patterns, and ultimately given rise to different social systems?

- What is the genetic architecture of baboon social behavior (including social style, patterns of dispersal, and degree of reproductive skew according to social status)? How does that architecture promote or restrict evolutionary flexibility in social systems?

- Does the link between sociality and reproductive success vary among species or even local populations?

- Do the different species vary in the way they represent the social relationships around them and how they attend to social information?

- How responsive are baboons to changes in temperature patterns due to global warming, as well as to associated changes in aridity or habitat type?

DOI: https://doi.org/10.7554/eLife.50989.005

## Baboons in the Anthropocene

Baboons allow us to study the effects of accelerating anthropogenic fragmentation, loss of natural habitats and climate change in a highly adaptable primate system. For example, baboons may rapidly change how long they allocate time and energy to different behaviors or where they range, in response to human-related activities and habitat changes (*Fehlmann et al., 2017*; *Ferreira da Silva et al., 2018*). Studies of individual baboon behavior can use sophisticated GPS tracking and non-invasive genetic tools to make broad-scale inferences about movements and processes at the population level (*Kopp et al., 2014*; *Strandburg-Peshkin et al., 2015*). These inferences can then be applied to questions of how other large populations of mammals will respond to changes in their environment.

Baboons are not considered a global priority in conservation, with the exception of Guinea baboons which are categorized as Near Threatened by the IUCN (*Oates et al., 2008*). However, populations geographically overlapping with human communities often damage crops and infrastructure and are persecuted as pests. In some locations, people consume substantial number of baboons and sell their meat in bushmeat markets (e.g., *Minhós et al., 2013*). Humans and baboons often compete for space and hunting of specific individuals or even entire groups is increasingly frequent, leading to fragmented populations and local extinctions (*Ferreira da Silva et al., 2018*). Non-monitored populations living outside protected areas may be at a higher risk of silently disappearing. The challenge is to assess the risk of different

populations and develop appropriate conservation plans.

The long-term nature of many baboon field studies has provided great insight into how populations may rise and fall rapidly in response to changes in the environment. The Amboseli Baboon Project, for example, has been running continuously for 50 years, and has documented numerous periods of relative drought or rainfall abundance (*Alberts et al., 2005*; *Alberts and Altmann, 2011*). This variation in precipitation has been linked to variation in fecundity and survival (*Beehner et al., 2006*; *Lea et al., 2015*) and to subsequent changes in population structure (*Altmann et al., 1985*). Periods of environmental change, and consequent boom and bust cycles in populations, are driven by both natural phenomena, such as natural aging of woodland, and anthropogenic influences, such as overgrazing by pastoralists (*Alberts and Altmann, 2011*). Many long-term baboon field sites also carefully collect detailed data on temperature and rainfall, as well as food availability and diseases. They also monitor the habitats in addition to the baboon populations. The breadth and scope of such data ensure that the baboon represents an outstanding model of both individual-level and population-level responses to environmental change.

## Conclusion

Baboons constitute a fascinating and informative analog model for hominin evolution in savanna habitats, with their ongoing patterns of range expansions and contractions, and regular occurrences of hybridization where two species meet. Given the availability of long-term data and the variation in the types of societies baboons live in, they constitute an excellent test case to study the link between sociality, health, longevity and reproductive success, as well as the emergence and spread of diseases. Such studies are extremely important to put biomedical data from captive baboon studies into natural context. For future research, we propose an approach that integrates field observations and carefully designed field experiments with cutting-edge measures of genomic variation, gene expression, non-invasive endocrinology and immunology. The fact that baboons have been studied in a wide range of habitats at sites across Africa for several decades also make them an informative example to investigate how populations of large mammals respond to environmental diversity and change (see *Box 2* for suggested future research questions).

## Acknowledgements
This paper is dedicated to the memory of Dorothy L Cheney who loved baboons. We thank the funders of the Frontiers in Baboon Research Symposium, namely the Deutsche Forschungsgemeinschaft (FI 707/21–1), the Leibniz Science-Campus Primate Cognition and the University of Göttingen. We thank PJ Perry and the editors and reviewers, for valuable comments and suggestions.

**Julia Fischer** is in the Cognitive Ethology Laboratory at the German Primate Center, Leibniz-Institute for Primate Research, the Department of Primate Cognition, Georg-August-University of Göttingen, and at the Leibniz Science Campus for Primate Cognition, Göttingen, Germany
jfischer@dpz.eu
https://orcid.org/0000-0002-5807-0074

**James P Higham** is in the Department of Anthropology, New York University and the New York Consortium in Evolutionary Primatology, New York, United States
jhigham@nyu.edu
https://orcid.org/0000-0002-1133-2030

**Susan C Alberts** in the Departments of Biology and Evolutionary Anthropology, Duke University, Durham, United States and at the Institute of Primate Research, Nairobi, Kenya
https://orcid.org/0000-0002-1313-488X

**Louise Barrett** is in the Department of Psychology, University of Lethbridge, Lethbridge, Canada and the Applied Behavioural Ecology and Ecosystems Research Unit, University of South Africa, Pretoria, South Africa

**Jacinta C Beehner** is in the Departments of Psychology and Anthropology, University of Michigan, Ann Arbor, United States

**Thore J Bergman** is in the Departments of Psychology and Anthropology, University of Michigan, Ann Arbor, United States

**Alecia J Carter** is at the Institut des Sciences de l'Evolution de Montpellier, Montpellier, France and Université de Montpellier, CNRS, IRD, EPHE, Montpellier, France
https://orcid.org/0000-0001-5550-9312

**Anthony Collins** is at the Gombe Stream Research Centre, Jane Goodall Institute, Kigoma, Tanzania

**Sarah Elton** is in the Department of Anthropology, Durham University, Durham, United Kingdom

**Joël Fagot** is at Aix Marseille Université, Marseille, France, and Centre Nationalde la Recherche Scientifique, Montpellier, France

**Maria J Ferreira da Silva** is in the Organisms and Environment Division, School of Biosciences, Cardiff University, Cardiff, United Kingdom; the Centro de

Investigação em Biodiversidade e Recursos Genéticos, Universidade do Porto, Portugal; and the Centro de Administração e Políticas Públicas, School of Social and Political Sciences, University of Lisbon, Lisbon, Portugal

**Kurt Hammerschmidt** in the Cognitive Ethology Laboratory at the German Primate Center, Leibniz-Institute for Primate Research, Göttingen, Germany

**Peter Henzi** is in the Department of Psychology, University of Lethbridge, Lethbridge, Canada and the Applied Behavioural Ecology and Ecosystems Research Unit, University of South Africa, Pretoria, South Africa

**Clifford Jolly** is in the Department of Anthropology, New York University and the New York Consortium in Evolutionary Primatology, New York, United States

**Sascha Knauf** is in the Work Group Neglected Tropical Diseases, Infection Biology Unit, German Primate Center, Leibniz-Institute for Primate Research, and the Division of Microbiology and Animal Hygiene, Georg-August-University, Göttingen, Germany
https://orcid.org/0000-0001-5744-4946

**Gisela H Kopp** is in the Zukunftskolleg, the Department of Biology and the Centre for the Advanced Study of Collective Behaviour at University of Konstanz, and the Department of Migration, Max Planck Institute for Animal Behaviour in Konstanz, Germany

**Jeffrey Rogers** is at the Human Genome Sequencing Center and the Department of Molecular and Human Genetics, Baylor College of Medicine, Houston, United States

**Christian Roos** is in the Gene Bank of Primates and the Primate Genetics Laboratory at the German Primate Center, Leibniz-Institute for Primate Research, Göttingen, Germany

**Caroline Ross** is in the Department of Life Sciences, Roehampton University, London, United Kingdom

**Robert M Seyfarth** is in the Department of Psychology, University of Pennsylvania, Philadelphia, United States

**Joan Silk** is in the School of Human Evolution and Social Change, and the Institute for Human Origins, Arizona State University, Tempe, United States

**Noah Snyder-Mackler** is in the Department of Psychology, the Center for Studies in Demography and Ecology and the National Primate Research Center at the University of Washington, Seattle, United States

**Veronika Städele** is at the Max Planck Institute for Evolutionary Anthropology, Leipzig, Germany

**Larissa Swedell** is in the Department of Anthropology, Queens College, City University of New York and the New York Consortium in Evolutionary Primatology, New York, United States and the Department of Archaeology, University of Cape Town, Cape Town, South Africa

**Michael L Wilson** is in the Departments of Anthropology and Ecology, Evolution and Behavior at University of Minnesota, Minneapolis, and the Institute on the Environment, University of Minnesota, Saint Paul, United States

**Dietmar Zinner** is in the Cognitive Ethology Laboratory at the German Primate Center, Leibniz-Institute for Primate Research and at the Leibniz ScienceCampus for Primate Cognition, Göttingen, Germany
https://orcid.org/0000-0003-3967-8014

*Author contributions:* Julia Fischer, James P Higham, Conceptualization, Visualization, Writing—original draft, Writing—review and editing; Susan C Alberts, Louise Barrett, Jacinta C Beehner, Thore J Bergman, Alecia J Carter, Anthony Collins, Sarah Elton, Joël Fagot, Maria Joana Ferreira da Silva, Kurt Hammersch-midt, Peter Henzi, Clifford J Jolly, Sascha Knauf, Gisela H Kopp, Jeffrey Rogers, Christian Roos, Caroline Ross, Robert M Seyfarth, Joan Silk, Noah Snyder-Mackler, Veronika Staedele, Larissa Swedell, Michael L Wilson, Writing—review and editing; Dietmar Zinner, Writing—original draft, Writing—review and editing

*Competing interests:* The authors declare that no competing interests exist.

## Funding

| Funder | Grant reference number | Author |
| --- | --- | --- |
| Deutsche Forschungsgemeinschaft | FI 707/21-1 | Julia Fischer |

The funders had no role in study design, data collection and interpretation, or the decision to submit the work for publication.

**Decision letter and Author response**
Decision letter https://doi.org/10.7554/eLife.50989.008
Author response https://doi.org/10.7554/eLife.50989.009

## Additional files

### Data availability

This is a review article; there are no data sets associated with this article.

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
