## [Decision Letter]

Thank you for submitting your article "The Natural History of Model Organisms: Insights into the evolution of social systems and species from baboon studies" for consideration by *eLife*. Your article has been reviewed by two peer reviewers, and the evaluation has been overseen by two Features Editors at *eLife* (Stuart King and Peter Rodgers). The following individual involved in review of your submission has agreed to reveal their identity: Jason Kamilar.

The reviewers have discussed the reviews with one another and the Associate Features Editor has drafted this decision to help you prepare a revised submission.

Summary:

This essay is being considered as part of a series of articles on "The Natural History of Model Organisms": https://elifesciences.org/collections/8de90445/the-natural-history-of-model-organisms. Each article should explain how our knowledge of the natural history of a model organism has informed recent advances in biology, and how understanding its natural history can influence/advance future studies.

Overall, it was a pleasure to read this clear summary of current knowledge about the natural history of baboons. The manuscript is a wide-ranging, authoritative, and interesting read. The authors have done an excellent job of concisely presenting an overview of baboon biology and its relevance for being considered a model organism in ecology and evolution. The reviewers expect this paper to be highly cited and look forward to seeing it published after a few revisions are made. The hope is that the revisions will provide some more context for readers who are less familiar with baboon research, and help condense the article slightly to make it more focused, which could help to increase its ultimate impact.

Essential revisions:

1) This manuscript could be considered to be an interesting outlier for the "Natural History of Model Organisms" series, which has thus far focused on species that have largely been studied in the lab or under domestication (or are close relatives of a lab model organism). In contrast, this paper focuses exclusively on research in the wild, on a set of species that are unusually well studied in their natural context. For the benefit of non-baboon researchers, it would be helpful if the Introduction acknowledged that baboons may seem to be an unusual choice for a model organism, but then explain what it is that makes them such a good model for certain questions (i.e. better than other species that are routinely studied in the lab). A mention of the rather extensive work on baboons in captivity, where they are among the most intensively studied non-human primates in biomedical science, would also be appreciated here. Work in the wild extends the scope of what the community can learn from baboons beyond captive research, and it would be interesting if the authors shared their thoughts as to whether it also helps to motivate the research done with captive baboons, or to interpret those results?

2) The sections on systematics and phylogeography could be combined with some text currently in the Introduction to give a more general introduction to the "Natural history of baboons". Restructuring this text may also present opportunities to cut some text, because, for example, geographic ranges are currently discussed in three separate places. Breaking the new section into short, clearly defined sub-sections (i.e. Systematics, Morphology, Habitats, Diet etc.) may also help the reader to navigate this background information.

3) The reviewers felt that the discussion of allele surfing (subsection "Population genetics and range expansion", third paragraph) could probably be removed. While this could be interesting and relevant to baboon evolution, none of the citations in this paragraph are actually about baboons. As such, it was unclear how baboons have contributed thus far to this question or even whether they would be better models than, say, humans or classical lab model organisms. Removing the text would help to keep the article focused.

4) Given that they touch upon related topics, the section on "Baboons in the Anthropocene" could be moved to follow the section on "Population and range expansion". This change could help this later section to feel more integrated in the article, and would mean that the article's structure more closely follows the order of the research areas that are very nicely articulated in the Conclusion.

5) The sections on "Variation in social organization and behavior" and "Previously understudied species reveal hidden diversity" should also be integrated into one and condensed, since the hidden diversity discussed is also about social organization and behavior. The Social Cognition section could also likely be condensed by a few sentences.

6) The section on "Functional genomics" seems to be more tool or method-driven that the previous sections and is almost all future-oriented. It would be good if it could give more concrete examples of how, when applied to baboons, these methods could advance broad biological knowledge. If the section on "Baboons in the Anthropocene" is moved earlier in the text, this section would be the penultimate in the article. As such, one approach would be to use this section to explain how/why functional genomic approaches could be leveraged to advance the questions laid out in the previous sections, especially if the authors can include examples from the recent literature. Alternatively, the authors should explicitly note that, compared to the other areas highlighted, there's much more to be done in this area of research.

7) Throughout the article there are a few places where the paper may assume too much about domain/disciplinary knowledge on the part of readers. The Glossary can help with this, if the definitions provided are written to be as accessible as possibly. Most importantly, the reviewers suggest explaining what "rank" is (it first appears in the subsection "Variation in social organization and behavior"), whether it's sex-specific, and why it's important in baboons. Other cases include the use of the term "despotic" (likely to be familiar to primatologists, but not to many others), mention of copulation calls (why is it meaningful that they might differ in chacma baboons?), and citing "the African multi-regionalism" hypothesis (a reader may know why it's important that baboons are a model for this hypothesis if they don't know what it is).

---

## [Author Response]

Essential revisions:1) This manuscript could be considered to be an interesting outlier for the "Natural History of Model Organisms" series, which has thus far focused on species that have largely been studied in the lab or under domestication (or are close relatives of a lab model organism). In contrast, this paper focuses exclusively on research in the wild, on a set of species that are unusually well studied in their natural context. For the benefit of non-baboon researchers, it would be helpful if the Introduction acknowledged that baboons may seem to be an unusual choice for a model organism, but then explain what it is that makes them such a good model for certain questions (i.e. better than other species that are routinely studied in the lab). A mention of the rather extensive work on baboons in captivity, where they are among the most intensively studied non-human primates in biomedical science, would also be appreciated here. Work in the wild extends the scope of what the community can learn from baboons beyond captive research, and it would be interesting if the authors shared their thoughts as to whether it also helps to motivate the research done with captive baboons, or to interpret those results?

We have followed this suggestion, and tried to be more explicit why this paper might be relevant for scientists who use baboons in biomedical research. In the Introduction, we now write: "In the 1950s, baboons became the subject of more systematic scientific enquiry, both in the field, and in captivity. […] An understanding of the evolution and natural life-history also allows assessment of the validity of results derived from captive populations." We also return to this theme briefly in the Conclusion.

2) The sections on systematics and phylogeography could be combined with some text currently in the Introduction to give a more general introduction to the "Natural history of baboons". Restructuring this text may also present opportunities to cut some text, because, for example, geographic ranges are currently discussed in three separate places. Breaking the new section into short, clearly defined sub-sections (i.e. Systematics, Morphology, Habitats, Diet etc.) may also help the reader to navigate this background information.

We thought a lot about this suggestion, and have partly implemented it. Our reasoning was that we first introduce the taxon and then talk about the different species. We feel that we need to introduce this variation first before we go on to present species differences in Morphology etc. There are only two sentences on Diet, so we did not establish a new subsection. We did, however, cut text in a number of places to avoid the highlighted redundancy.

3) The reviewers felt that the discussion of allele surfing (subsection "Population genetics and range expansion", third paragraph) could probably be removed. While this could be interesting and relevant to baboon evolution, none of the citations in this paragraph are actually about baboons. As such, it was unclear how baboons have contributed thus far to this question or even whether they would be better models than, say, humans or classical lab model organisms. Removing the text would help to keep the article focused.

We agree, and have removed this section.

4) Given that they touch upon related topics, the section on "Baboons in the Anthropocene" could be moved to follow the section on "Population and range expansion". This change could help this later section to feel more integrated in the article, and would mean that the article's structure more closely follows the order of the research areas that are very nicely articulated in the Conclusion.

We understand this suggestion, but we believe that the "Baboons in the Anthropocene" section belongs as the final section, as it is prospective, and about the future of baboons. We have, however, edited the Conclusion such that the order there accurately reflects the order of the content.

5) The sections on "Variation in social organization and behavior" and "Previously understudied species reveal hidden diversity" should also be integrated into one and condensed, since the hidden diversity discussed is also about social organization and behavior. The Social Cognition section could also likely be condensed by a few sentences.

We have followed this suggestion and have both integrated and shortened these two sections.

6) The section on "Functional genomics" seems to be more tool or method-driven that the previous sections and is almost all future-oriented. It would be good if it could give more concrete examples of how, when applied to baboons, these methods could advance broad biological knowledge. If the section on "Baboons in the Anthropocene" is moved earlier in the text, this section would be the penultimate in the article. As such, one approach would be to use this section to explain how/why functional genomic approaches could be leveraged to advance the questions laid out in the previous sections, especially if the authors can include examples from the recent literature. Alternatively, the authors should explicitly note that, compared to the other areas highlighted, there's much more to be done in this area of research.

Thank you for this suggestion. We now explicitly state that this is a research area on the cusp of blooming; we now state: "Researchers now have access to whole genome sequence data and other genomic information such as annotations for protein coding genes and transcriptome information (Robinson et al., 2019; Rogers et al., 2019; Vilgalys et al., 2019). We therefore expect an exponential increase in the number and diversity of available genomes, which will facilitate the genomic investigation of the molecular and cellular basis of baboon adaptations and adaptive flexibility."

7) Throughout the article there are a few places where the paper may assume too much about domain/disciplinary knowledge on the part of readers. The Glossary can help with this, if the definitions provided are written to be as accessible as possibly. Most importantly, the reviewers suggest explaining what "rank" is (it first appears in the subsection "Variation in social organization and behavior"), whether it's sex-specific, and why it's important in baboons. Other cases include the use of the term "despotic" (likely to be familiar to primatologists, but not to many others), mention of copulation calls (why is it meaningful that they might differ in chacma baboons?), and citing "the African multi-regionalism" hypothesis (a reader may know why it's important that baboons are a model for this hypothesis if they don't know what it is).

We are sorry that we sometimes used too much jargon. We now explain the concept of rank and why it is important. We also explain despotism, and have used more general terms whenever we felt we could (e.g., perineal  anogenital; shorter periods of post-partum amenorrhea  shorter periods before they resume menstrual cycling following birth). We cut the African multi-regionalism hypothesis and simply refer to different hypotheses regarding human origins. We appreciate any further suggestions where we might remove further jargon that we may have overlooked.